# JustLogic: A Comprehensive Benchmark for Evaluating Deductive Reasoning in LLMs

Michael K. Chen*        Xikun Zhang        Dacheng Tao

Nanyang Technological University
Singapore

## Abstract

Logical reasoning is a critical component of Large Language Models (LLMs), and substantial research efforts in recent years have aimed to enhance their deductive reasoning abilities. However, existing deductive reasoning benchmarks, which are crucial for evaluating and advancing LLMs, suffer from significant constraints that restrict their utility, i.e., the lack of task complexity, the presence of prior knowledge as a confounder, and superficial error analysis. To address these deficiencies, we introduce JustLogic, a synthetically generated benchmark designed for rigorous evaluation of LLMs. JustLogic is (i) highly complex, capable of generating a diverse range of linguistic patterns, vocabulary, and argument structures; (ii) prior knowledge independent, eliminating the advantage of models possessing prior knowledge and ensuring that only deductive reasoning is used to answer questions; and (iii) capable of in-depth error analysis on the heterogeneous effects of reasoning depth and argument form on model accuracy. Our experimental results on JustLogic reveal that (i) state-of-the-art (SOTA) reasoning LLMs perform on par or better than the human average but significantly worse than the human ceiling, and (ii) SOTA non-reasoning models still underperform the human average. All code and data are available at https://github.com/michaelchen-lab/JustLogic

## 1   Introduction

Deductive reasoning is a crucial capability for large language models (LLMs). It refers to the process of creating logically valid arguments, where conclusions necessarily follow from the premises. In other words, if an argument's premises are true, the conclusion must also be true. Recent state-of-the-art (SOTA) LLMs [1; 10; 15] have exhibited outstanding performance and consistent improvement across various reasoning benchmarks, including HelloSwag [34], ARC Challenge [6] and WinoGrande [22]. However, we argue that the existing benchmarks are insufficient and often ineffective for evaluating LLMs' true deductive reasoning capabilities.

We identify three major problems with the existing benchmarks. **First**, they lack complexity, as measured on two dimensions: natural language complexity, i.e. how arguments are linguistically expressed, and argument complexity, i.e. the structure of the argument itself. Manually curated datasets, such as FOLIO [12] and LogiQA 2.0 [21; 19] exhibit high natural language complexity but low argument complexity, while synthetic datasets like CLUTRR [24] and ProofWriter [26] exhibit the opposite. Simplicity in either dimension makes these benchmarks prone to overfitting and memorization, thus allowing models to perform well despite underlying weaknesses in logical reasoning. A more detailed analysis can be found in Section 3.4. **Second**, existing benchmarks often fail to test deductive reasoning in isolation, as models can benefit from prior knowledge. To empirically validate this claim, we developed a novel test for prior knowledge independence,

---

*Correspondence to: michaelchenkj@gmail.com, xikun.zhang@ntu.edu.sg, and dacheng.tao@ntu.edu.sg

Submitted to 39th Conference on Neural Information Processing Systems (NeurIPS 2025). Do not distribute.

which measures the influence of prior knowledge on reasoning benchmarks. As detailed in Section 5.1, prior knowledge can substantially increase accuracy, even in datasets not intended to require commonsense or domain knowledge, e.g. FOLIO and LogiQA 2.0. Thus, high accuracy may not reflect strong reasoning capabilities. **Third**, many existing benchmarks provide superficial error analysis, leaving key questions unanswered: At what reasoning depth does the model start to fail? How does the model compare to humans at different argument depths? Which argument forms is the model particularly weak at? These insights are essential for understanding the depth and robustness of a model's deductive reasoning, yet not many benchmarks provide them. Section 5.3 demonstrates the importance and usefulness of comprehensive error analysis.

---

**Paragraph:**

- It is a fact that either relics are artifacts or dogs are capable of barking. [$p \lor q$]
- If relics are artifacts, then fertilizers contain phosphorus. [$p \to r$]
- Assuming dogs are capable of barking, we know that fertilizers contain phosphorus. [$q \to r$]

**Statement:** Fertilizers contain phosphorus. [$r$]
**Question:** Is the statement true, false, or uncertain?
**Answer:** True

---

Figure 1: Example of a question adapted using JustLogic's dataset construction algorithm. Note that the statements are *intentionally* factually inaccurate, which we explain and justify in Section 3.2. Formal notations are included for illustrative purposes and are not provided to models.

Due to these issues, LLMs' deductive reasoning abilities remain ambiguous. In response to the critical need for a reliable benchmark to support ongoing research efforts, we present JustLogic, a novel natural language deductive reasoning benchmark. The task is to determine whether a given statement is true, false, or uncertain, using only the given premises, which are assumed to be true. An example is shown in Figure 1.

JustLogic's construction ensures it is (i) complex, (ii) prior knowledge independent, and (iii) capable of in-depth error analysis. **First**, to achieve high argument and natural language complexity, JustLogic is code-generated rather than manually curated. This allows the generation of a theoretically infinite number of unique argument structures. Natural language sentences are then drawn from GenericsKB-Best [4], a database of 1M+ unique real-world sentences, and inserted into the argument structures, introducing high natural language complexity. **Second**, since sentences are randomly sampled from the entire GenericsKB-Best dataset, the generated arguments generally do not align with real-world knowledge, thereby eliminating prior knowledge as a confounder. **Finally**, in-depth error analysis is enabled by the programmatic generation process, which enables the inspection of each question's properties, such as reasoning depth and argument form, to investigate their impact on model performance. A comparison between JustLogic and four similar logical reasoning benchmarks (CLUTRR, ProofWriter, LogiQA 2.0, and FOLIO) is presented in Table 1, with further details on dataset construction provided in Section 3.

Using JustLogic, we conducted comprehensive experiments to evaluate the deductive reasoning capabilities of current LLMs. First, our novel prior knowledge independence test demonstrated that prior knowledge enables LLMs to bypass deductive reasoning on existing datasets, resulting in artificially high accuracies. This is not observed in JustLogic. Second, we benchmarked the performance of SOTA LLMs and human participants using JustLogic. Most SOTA LLMs performed significantly lower than the average human accuracy (73.0%). Only DeepSeek R1 performed substantially better (80.9%), but still fell short of the human ceiling (100.0%). Finally, enabled by JustLogic's code-generated nature, our thorough error analysis examined the heterogeneous impact of argument structure and reasoning depth on model performance. These experiments show that the JustLogic benchmark is a reliable test of deductive reasoning and reveals significant room for improvement in LLMs.

In summary, our contributions are threefold. First, we evaluate the limitations of existing benchmarks. Second, we introduce the JustLogic benchmark, a synthetic dataset to evaluate deductive reasoning, that addresses the aforementioned limitations. Third, our experiments using JustLogic demonstrate

Table 1: Comparison of JustLogic with other deductive reasoning datasets. The symbol ∼ suggests the feature is present but to a limited extent.

| | High NL Complexity | High Arg. Complexity | Prior Knowledge Independence | In-Depth Error Analysis |
|---|---|---|---|---|
| CLUTRR | ✗ | ✓ | ✓ | ✓ |
| ProofWriter | ✗ | ✓ | ✓ | ∼ |
| LogiQA 2.0 | ✓ | ✗ | ✗ | ∼ |
| FOLIO | ✓ | ✗ | ✗ | ∼ |
| **JustLogic** | ✓ | ✓ | ✓ | ✓ |

that most SOTA models perform significantly worse than humans. We posit that the deductive reasoning capabilities of LLMs still have significant room for improvement, and hope that the JustLogic benchmark will assist researchers in designing and evaluating LLMs.

## 2 Related Work

### 2.1 Existing reasoning datasets for Large Language Models

Reasoning benchmarks are a vital part of LLM evaluation. Some benchmarks measure deductive reasoning in conjunction with natural language inference (NLI), inductive reasoning, and common-sense knowledge: HellaSwag [34] tasks machines to select the most likely follow-up of an event description, WinoGrande [22] is a pronoun resolution task, and MuSR [25] tasks machines to solve fictional problems, such as murder mysteries. Other benchmarks measure reasoning on domain knowledge: AI2 Reasoning Challenge (ARC) [31] contains grade-school science questions, while Massive Multitask Language Understanding (MMLU) [13] contains questions across 57 subjects in STEM, humanities, and more. Finally, math-specific benchmarks include GSM-8K [7] and DROP [9].

The aforementioned benchmarks explicitly evaluate skills beyond reasoning and do not specifically define the type of reasoning involved, e.g. inductive, deductive, and analogical. As such, benchmarks that solely test for deductive reasoning have seen a considerable increase in interest. They can be classified into two broad categories: synthetic and manually curated. Synthetically-generated datasets include (i) CLUTRR [24], where a machine must infer the relationship of two family members based on stories, (ii) ProofWriter [26], where a machine must deduce a statement's truth value based on a set of facts and rules, and (iii) ProntoQA-OOD [23], where a machine must prove a statement based on a set of facts. Manually curated datasets include (i) LogiQA 2.0 [19], containing manually-translated logical reasoning questions from the Chinese Civil Service Exam, (ii) FOLIO [12], containing questions with manually-annotated content using Wikipedia pages, and (iii) ReClor [33], containing reading comprehension questions from GMAT and LSAT.

As discussed earlier, synthetic datasets are prior knowledge independent and exhibit high argument and low natural language complexity; manually curated datasets are the opposite. JustLogic, being synthetic, contains all its advantages while offering the natural language complexity of manually curated datasets, which we further explained in Section 3.4 and 5.1.

### 2.2 Reasoning in Large Language Models

As LLMs continue to increase in size, their performance on various reasoning-related benchmarks has improved dramatically. For example, in 2024, Gemini Ultra [27] achieved 90.0% on MMLU when the SOTA model in 2020, UnifiedQA 11B [17], achieved a mere 48.9%. In 2023, GPT-4 achieved 96.4% on ARC when the SOTA model in 2020, GPT-3 [5], achieved 53.2%.

The advent of prompting techniques played an important role in developing LLMs' reasoning abilities. In-context learning [8] provides LLMs with instructions and examples in the input prompt to guide its response. Chain-of-thought prompting [29] prompts LLMs to generate a series of intermediate reasoning steps before arriving at the final answer. Self-consistency decoding [28] chooses the most consistent answer after sampling multiple chain-of-thought outputs. Least-to-most prompting [36] decomposes a complex problem into simpler subproblems, which are then solved sequentially.

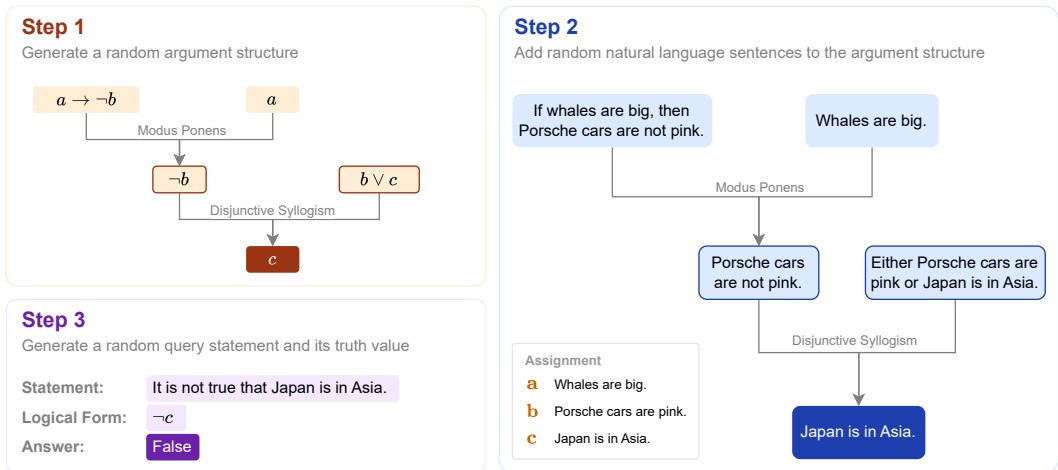

Figure 2: A step-by-step example of how an instance with a reasoning depth of 2 is constructed.

As mentioned above, LLMs are conventionally tested on datasets that combine reasoning with other skills. Moreover, existing logical reasoning-specific datasets possess major limitations that call into question the reliability of their evaluations. In response, JustLogic aims to robustly and accurately evaluate the deductive reasoning abilities of LLMs.

# 3 Dataset Construction

JustLogic is a programmatically generated dataset designed to evaluate a model's ability of deductive reasoning, specifically its capability to form logically valid arguments. A logically valid argument is one where the conclusion necessarily follows from the premise(s); in other words, given the premises are true, the conclusion must also be true.

In order to test this, JustLogic presents a model with a paragraph consisting of premises, followed by a query statement. Based solely on the premises and assuming they are all true, the model needs to determine whether the query statement is true, false, or uncertain. In line with the open-world assumption, the "Uncertain" answer refers to cases where the premises neither support nor contradict the query statement.

The following outlines the process for generating each instance in the dataset, which is exemplified by Figure 2.

1. Step 1: Generate an argument structure.
2. Step 2: Add natural language statements to the argument structure.
3. Step 3: Generate a query statement.

## 3.1 Step 1: Generate argument structure

Argument structures are composed of one or more valid argument forms, derived from propositional logic; argument forms are made up of a series of logical forms, which we define as symbolic representations of statements. Specifically, the seven distinct argument forms in our dataset are constructed with the following four logical forms: (i) basic ($x$), (ii) negation ($\neg x$), (iii) conditional ($x \rightarrow y$), and (iv) disjunction ($x \vee y$). While there is a theoretically infinite number of possible argument forms, complex argument forms can be derived by combining simpler ones. Therefore, we explicitly define the seven most fundamental forms [16]: modus ponens, modus tollens, hypothetical syllogism, disjunctive syllogism, reductio ad absurdum, constructive dilemma, and disjunctive elimination. Table 6 in Appendix A provides the corresponding formal notations and natural language examples.

The algorithm to create an argument structure (formally shown in Appendix B) accepts an intended argument depth as input. It first generates a random conclusion and an argument form to support it, which in Figure 2 is $c$ and disjunctive syllogism. If the intended depth has not been reached, one

Table 2: Expressions of logical forms.

|  | Formal Notation | Sample Expression | No. of Expr. |
|---|---|---|---|
| Basic | $x$ | The claim that $x$ holds true. | 16 |
| Negation | $\neg x$ | The claim that $x$ does not reflect reality. | 15 |
| Conditional | $x \rightarrow y$ | Once we know that $x$, we also know that $y$. | 11 |
| Disjunction | $x \vee y$ | It is a fact that either $x$ or $y$. | 8 |

or more premises will become subconclusions, which are supported by new, randomly generated argument forms. In Figure 2, this is exemplified by $\neg b$ becoming a subconclusion, supported by a modus ponens argument. This process continues until the desired depth is achieved.

### 3.2 Step 2: Adding natural language

Next, the symbolic statements are converted into natural language. Each statement consists of one or more logical forms, *i.e.* variable, negation, conditional, and disjunction. In natural language, these forms can be expressed in a variety of ways. For example, a conditional can be expressed as both "If $x$, then $y$." and "Given that $x$, $y$ is true.", where variables $x$ and $y$ are simple propositions. To emulate the diversity of natural language, a list of expressions for each logical form is created using human feedback, and potentially aided by LLMs. Table 2 shows the formal notation of each form, alongside a sample expression and the total number of unique expressions. The variable(s) within each expression are ultimately replaced by randomly selected generic, real-world sentences from GenericsKB-Best [4]. The GenericsKB-Best dataset is chosen for its vast collection of simple propositions (1,020,868 sentences) without conditionals, disjunctions, etc. A complete example can be found in Step 2 of Figure 2.

Notably, as shown in Figure 2, the statements are generally factually inaccurate despite being drawn from real-world data. This is intentional. Real-world propositions allow us to generate sentences with diverse grammatical structures that closely emulate human-written arguments. However, factually accurate arguments enable models to bypass deductive reasoning with their prior real-world knowledge, which is experimentally demonstrated in Section 5.1. By using real-world yet factually inaccurate statements, we combine realism and prior knowledge independence.

There are potential concerns that factually inaccurate statements and "unnatural" synthetic language may confuse models and lead to artificially low performance. Appendix F.2 and F.3 empirically refute these concerns.

### 3.3 Step 3: Generate query statement

The LLM's task is to determine whether the given query statement is true, false, or uncertain based on the premises provided. Using Figure 2 as an example, if we assign the query statement to be the negation of the conclusion, i.e. "It is not true that Japan is in Asia", then the answer is false. If the query statement is the same as the conclusion, then the answer is true. If the query statement is unrelated to the premises, then the answer is uncertain.

### 3.4 Dataset Complexity

In the context of deductive reasoning datasets, complexity is defined as the variety and comprehensiveness of instances. It can be further divided into two dimensions: natural language complexity and argument complexity. In this section, we highlight the significance of both aspects and how JustLogic compares against other logical reasoning datasets.

**Natural language complexity.** Human language is complex. Statements and arguments of similar meanings can be presented in a variety of ways. Therefore, it is insufficient for models to reason solely with symbols, *e.g.* $x$ and $y$, and basic natural language sentences, *e.g.* "Some birds are yellow."; they must be capable of reasoning with real-world vocabulary and diverse sentence structures to be useful in practical contexts.

We measure natural language complexity with (i) reading difficulty, as measured by the Flesch-Kincaid Grade Level test [18], and (ii) lexical diversity, as measured by vocabulary & domain size.

Table 3: Statistics of dataset complexity.

| | Natural Language | | Argument | |
|---|---|---|---|---|
| | Reading Difficulty ↑ | Vocab. (Domains) | Reasoning Depth | Arg. Structures |
| CLUTRR | 6.67 | 1396 (1) | $1 - \infty$ | $\infty$ |
| ProofWriter | 0.96 | 101 (×) | $1 - \infty$ | $\infty$ |
| LogiQA 2.0 | 17.10 | 10004 (>10) | × | × |
| FOLIO | 18.75 | 4351 (>10) | 1 - 7 | 76 |
| **JustLogic** | **20.55** | **10557 (>10)** | **$1 - \infty$** | $\infty$ |

For (i), the score is presented as a U.S. grade level; the higher the score, the harder the text is to read. Scores greater than 12 should be used to compare the relative difficulty between benchmarks, with higher scores indicating relatively greater textual complexity. A domain is defined as any topic of interest, such as golf, computers, or traveling; Vocabulary size refers to the number of unique words in the dataset. Given the difficulty of quantitatively capturing linguistic complexity, Appendix C also shows text samples of each benchmark, representative of their complexity.

As shown in Table 3, existing synthetic datasets have low natural language complexity, while human-written datasets, such as FOLIO and LogiQA 2.0, exhibit significantly higher complexity. This is expected since synthetic datasets translate symbols in formal logic into natural language using limited templates of sentence structures and vocabulary lists. For example, in ProofWriter, a typical sentence follows the format "All dogs are (not) red.". The linguistic patterns of human-written datasets, in contrast, are bound only by human creativity. Despite being synthetic, JustLogic, achieves natural language complexity on par with manually curated datasets, due to its comprehensive selection of expressions and the use of GenericsKB-Best as the source of sentences.

**Argument complexity.** Argument complexity refers to the diversity of argument structures used in the dataset. A sufficiently high argument complexity is important because humans use a range of argument forms to reason, beyond just conditionals and modus ponens. Moreover, a real-world argument is typically composed of multiple argument forms, due to their inherent complexity.

A dataset's argument complexity is evaluated based on two metrics: (i) range of reasoning depth, and (ii) number of unique argument structures. The upper limit of both metrics is calculated based on the theoretical maximum without any additional human input, rather than the highest depth used in experiments in existing works. For example, CLUTRR's dataset construction program can generate any number of depths (referred to as relation length in the original paper), despite its experiments only utilizing questions of up to a depth of 10. Thus, its upper limit of depth is infinite.

Table 3 shows that synthetic datasets, such as CLUTRR, ProofWriter, and JustLogic, excel in this area, as there is no upper limit to their reasoning depth and number of argument structures. Manually curated datasets, in contrast, either lack an explicit concept of reasoning depth and argument structures (e.g. LogiQA 2.0), or have a limited selection of both (e.g. FOLIO). While manual datasets require significant human efforts and investment to expand their complexity, synthetic ones can scale trivially.

In summary, JustLogic combines the best of both dataset construction methods: the argument complexity of synthetic datasets and the natural language complexity of manually curated ones.

### 3.5 Future-proofing JustLogic

As the reasoning abilities of LLMs continue to improve, we expect LLMs to solve the existing JustLogic dataset eventually. To maintain JustLogic's relevance, we leverage its synthetic nature to increase complexity with minimal human input. Argument complexity can be adjusted by increasing the (i) range of argument depth (empirically validated in Appendix H.3) and (ii) number of distinct argument forms to >7. Natural language complexity can be adjusted by (i) increasing the number of expressions for each logical form and (ii) integrating more complex knowledge bases than GenericsKB. Importantly, these changes are programmatically achievable with minimal man-hours.

Importantly, JustLogic can also effectively tackle benchmark leakage [30], whereby test sets are unintentionally included in LLMs' pertaining data, thus artificially inflating their performance through memorization. Should JustLogic's test set be leaked, a new test set of similar difficulty can be trivially generated, thereby mitigating this problem.

## 4  Experimental Setup

We first investigate the influence of prior knowledge on evaluating deductive reasoning with JustLogic and other existing benchmarks using our prior knowledge independence test. Next, several SOTA LLMs of various sizes are evaluated using JustLogic. Finally, an in-depth error analysis of the LLMs' results is conducted.

JustLogic contains 7000 instances, equally split amongst reasoning depths ranging from 1 to 7. The test set (15% of all instances) is used for evaluation. Note that the number of instances and range of reasoning depths can be easily adjusted using JustLogic's open-source dataset generation program.

### 4.1  Prior Knowledge Independence Test

The task for deductive reasoning benchmarks is typically framed as $CQO \rightarrow A$: Given a context $C$, consisting of $n$ premises ($P = \{p_1, p_2, ..., p_n\}$), a question $Q$, and $m$ answer options ($O = \{o_1, o_2, ..., o_m\}$), determine the correct answer $A$. To assess the influence of prior knowledge on determining answer $A$, the prior knowledge independence test is framed as $QO \rightarrow A$. No context $C$ is provided, and the prompt instructs the LLM to answer the question based on prior knowledge alone. An example is provided in Appendix D.

If prior knowledge is not useful, the LLM should be unable to answer question $Q$ without $C$, and the accuracy for the prior knowledge independence test should approximate random probability $\frac{1}{m}$. Benchmarks exhibiting such accuracies are deemed prior knowledge independent.

While any LLM will be suitable, we use GPT-4 for its extensive prior knowledge. Similar results are replicated using Llama3-70B in Appendix F.1. The test is conducted on JustLogic and other logical reasoning benchmarks, i.e. CLUTRR, ProofWriter, LogiQA 2.0, and FOLIO.

### 4.2  Evaluation of LLMs' Deductive Reasoning

Our task, as illustrated in Figure 1, follows the conventional formulation: $CQO \rightarrow A$. Question $Q$ is "Is the statement $S$ true, false, or uncertain?"; there are 3 answer options, where $O = \{\text{true}, \text{false}, \text{uncertain}\}$. Prompts begin with a preamble, providing (i) the task instructions, (ii) a list of argument forms in propositional logic, and (iii) the available answer options.

We evaluated both reasoning and non-reasoning models of different sizes: Llama3-8B-Instruct [10], Llama3-70B-Instruct, GPT-4o-mini-2024-07-18, GPT-4o-2024-05-13, DeepSeek R1 Distill Qwen 14B, DeepSeek R1 [11], OpenAI o1-mini-2024-07-18, and OpenAI o1-2024-12-17 [14]. A range of prompting techniques are tested: zero-shot, few-shot, and chain-of-thought (CoT) [29]; more techniques are tested in Appendix H.1. Due to limited model access, 70 random samples are used for OpenAI o1 and 350 for the other reasoning models. To ensure fairness, the selected subset has the same proportion of each reasoning depth as the entire test set. Further implementation details are provided in Appendix E. Human performance is also measured; the experiment settings are detailed in Appendix G.

Finally, we perform an error analysis of the results from the aforementioned experiments, specifically examining the heterogeneous effects of argument form and reasoning depth on model accuracy. Accuracy for each argument form is only measured using questions with a reasoning depth of 1 since those with a depth of >1 typically have >1 argument forms. Lastly, a qualitative analysis of failure modes is conducted.

## 5  Results

### 5.1  Prior Knowledge Independence Test

The results of JustLogic and four other benchmarks are shown in Table 4; note that lower accuracy relative to the benchmark's random probability indicates that prior knowledge is more detrimental to answering the question, thereby demonstrating that the benchmark is more prior knowledge independent. Thus, the smaller the $|\Delta|$ between model accuracy and random probability, the better. The $|\Delta|$s of CLUTRR and ProofWriter are relatively low, while those of LogiQA 2.0 and FOLIO are nontrivially higher. This is because the former are synthetic datasets, while the latter are manually

curated. When a question is code-generated, it generally bears no correlation with reality, e.g. "Is it true, false, or uncertain that Gary is not red." from ProofWriter. Such questions are only answerable by reasoning over the context $C$. LogiQA 2.0 and FOLIO, on the other hand, often contain questions that are answerable purely using background knowledge, such as "Did the United States won the most medals in the last summer Olympic games?" from FOLIO. We posit that this is an unintentional consequence of the human bias to align the question's truth value with reality. While human curation enhances the question's realism, it compromises the test for deductive reasoning.

Table 4: Results of Prior Knowledge Independence Test. **The lower the $|\Delta|$, the better.**

|  | $|\Delta| \downarrow$ | Accuracy (%) | Random (%) |
|---|---|---|---|
| CLUTRR | 2.0 | 8.3 | 6.3 |
| ProofWriter | 3.7 | 37.0 | 33.3 |
| LogiQA 2.0 | 27.1 | 52.1 | 25.0 |
| FOLIO | 6.7 | 40.0 | 33.3 |
| JustLogic | **0.4** | 33.7 | 33.3 |

The JustLogic benchmark's $|\Delta|$ is 0.4%, given an accuracy of (33.7%) and random probability (33.3%), which is much lower than other benchmarks, including synthetic ones. The reason for this is twofold: first, JustLogic is also a synthetic dataset, which eliminates the human bias present in manually curated datasets. Second, while JustLogic uses real-world statements, their truth value is nonetheless randomly determined. For example, the statement "doors are solids" is factually true. However, by deducing from the paragraph, the correct answer is "False". Thus, using prior knowledge for many questions is not only unhelpful but also meaningfully decreases accuracy.

## 5.2 Evaluation of LLMs' Deductive Reasoning

As shown in Table 9, the best-performing model by a large margin is DeepSeek R1 with an accuracy of 80.9%. Surprisingly, OpenAI o1 (72.9%) performs substantially worse than DeepSeek R1; we provide deeper analysis on this in Appendix H.2. Several general observations can be made from these results: first, while models with larger parameter sizes generally perform better, they offer diminishing returns, shown by the accuracy gain of just 1.0% from Llama3-70B to GPT-4o, with both using CoT prompting. Second, the improvements offered by increasing model size pale in comparison to those offered by better prompting methods. Using CoT, Llama3-8B achieved higher performance (57.8%) than zero-shot Llama3-70B (53.1%). Lastly, reasoning models generally perform better given similar model sizes: the best reasoning model (DeepSeek R1) scored 15.3% higher than the best non-reasoning one (GPT-4o).

Table 5: Model and Human Evaluation Results.

|  | **0-shot** | **Few-shot** | **CoT** |
|---|---|---|---|
| Random Probability | 33.3 | 33.3 | 33.3 |
| Llama3-8B-Instruct | 49.8 | 41.8 | 57.8 |
| Llama3-70B-Instruct | 53.1 | 57.8 | 64.6 |
| GPT-4o-mini | 53.0 | 54.7 | 51.8 |
| GPT-4o | 53.8 | 58.3 | **65.6** |
| OpenAI o1-mini | - | - | 62.0 |
| OpenAI o1 | - | - | 72.9 |
| Qwen 14B (R1 Distill) | - | - | 61.7 |
| DeepSeek R1 | - | - | **80.9** |
| Human Average | 73.0 | 73.0 | 73.0 |
| Human Ceiling | 100.0 | 100.0 | **100.0** |

Human performance (73.0%) is higher than all models besides DeepSeek R1, while the human ceiling (100.0%) outperforms all models. The non-trivial gap between the human ceiling and the best-performing model (80.9%) shows that models still have significant room for improvement. Moreover,

we believe actual human performance might be higher than 73.0%. Given the long paragraphs of questions with high reasoning depth, participants may have predicted answers by briefly scanning the paragraph, rather than carefully deducing based on all available premises. This is supported by the suspiciously short time taken to complete the questions of some participants.

## 5.3 Error Analysis

Figure 3 illustrates the model accuracy by argument form (left) and by reasoning depth (right). The statistics of the best models of their respective categories are chosen: (i) Llama3-8B (small, non-reasoning model), (ii) Llama3-70B (large, non-reasoning model), (iii) OpenAI o1-mini (small, reasoning model), and (iv) DeepSeek (large, reasoning model). To mitigate noise, especially for models tested on smaller sample sizes, depths are grouped into low (1–3), medium (4–5), and high (6–7) categories. The qualitative analysis of failure modes can be found in Appendix I.

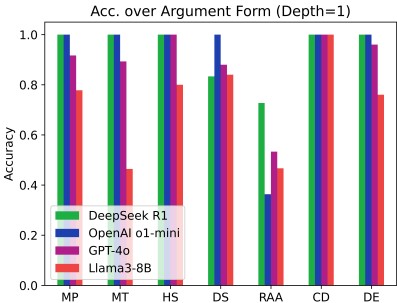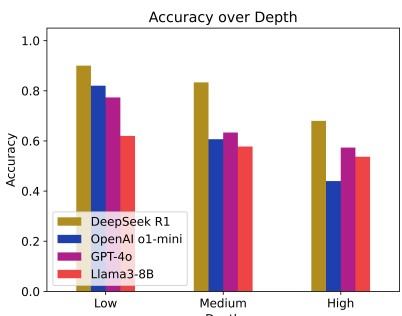

Figure 3: How argument form[2] and reasoning depth affects accuracy for various models.

While the relative accuracies of argument forms are heterogeneous across models, some forms perform distinctly better than others. For example, hypothetical syllogism and constructive dilemma achieve considerably higher performance than modus tollens, disjunctive syllogism, and reductio ad absurdum. Interestingly, the former argument forms are more commonly used by humans than the latter ones, which potentially hints at the cause of this observation.

As for reasoning depth, model accuracies generally decrease as depth increases, consistent with expectations that accuracy declines as the complexity of questions increases. Interestingly, OpenAI o1-mini performs comparably to DeepSeek R1 at low depths, but o1-mini sees a sharp decline in performance once depth increases, while DeepSeek R1 only sees a moderate decline; DeepSeek R1's superior performance is a result of better reasoning at higher reasoning depths. In fact, at medium and high depths, o1-mini no longer performs better than non-reasoning models, i.e. GPT-4o and Llama3-8B. These observations suggest that DeepSeek R1 supports deeper and longer lines of reasoning, which is crucial for deductive reasoning, and that large reasoning models perform drastically better than smaller ones on reasoning problems.

## 6 Conclusion

Deductive reasoning is one of the key challenges in LLM research. In response to the lack of reliable benchmarks, we present JustLogic, a natural language deductive reasoning dataset that is (i) highly complex, (ii) prior knowledge independent, and (iii) capable of in-depth error analysis. These qualities are enabled by JustLogic's dataset construction method: argument structures are synthetically generated, and natural language is programmatically incorporated via expression templates and a knowledge base. We empirically justify JustLogic's merits: most LLMs underperform the human average and all significantly underperform the human ceiling. We demonstrate that JustLogic is a highly challenging, future-proof benchmark that is reliable and insightful for evaluating logical reasoning in LLMs.

---

[2]MP = Modus Ponens, MT = Modus Tollens, HS = Hypothetical Syllogism, DS = Disjunctive Syllogism, RAA = Reductio Ad Absurdum, CD = Constructive Dilemma, DE = Disjunction Elimination

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

# A  Argument Forms

Table 6: An overview of the argument forms in the JustLogic dataset.

|  | Formal Notation | Example |
|---|---|---|
| Modus Ponens | $p \rightarrow q$ | If the sky is blue, then the dog is happy. |
|  | $p$ | The sky is blue. |
|  | $\vdash q$ | Therefore, the dog is happy. |
| Modus Tollens | $p \rightarrow q$ | If the sky is blue, then the dog is happy. |
|  | $\neg q$ | The dog is not happy. |
|  | $\vdash \neg p$ | Therefore, the sky is not blue. |
| Hypothetical Syllogism | $p \rightarrow q$ | If the sky is blue, then the dog is happy. |
|  | $q \rightarrow r$ | If the dog is happy, the owner is happy. |
|  | $\vdash p \rightarrow r$ | Therefore, the owner is happy. |
| Disjunctive Syllogism | $p \vee q$ | Either the dog is barking or the dog is asleep. |
|  | $\neg p$ | The dog is not barking. |
|  | $\vdash q$ | Therefore, the dog is asleep. |
| Reductio ad absurdum | $p \rightarrow q$ | If the dog is calm, the owner is around. |
|  | $p \rightarrow \neg q$ | If the dog is calm, the owner is not around. |
|  | $\vdash \neg p$ | Therefore, the dog is not calm. |
| Constructive Dilemma | $p \vee q$ | Either the sky is blue or it is raining. |
|  | $p \rightarrow r$ | If the sky is blue, the race can start. |
|  | $q \rightarrow s$ | If it is raining, the race is delayed. |
|  | $\vdash r \vee s$ | Therefore, either the race can start or it is delayed. |
| Disjunction Elimination | $p \vee q$ | Either the sky is blue or it is raining. |
|  | $p \rightarrow r$ | If the sky is blue, the dog is cheerful. |
|  | $q \rightarrow r$ | If it is raining, the dog is cheerful. |
|  | $\vdash r$ | Therefore, the dog is cheerful. |

# B  Algorithm for Step 1 of JustLogic's Dataset Construction

**Algorithm 1** Pseudocode for Step 1 (Generate argument structure) of JustLogic's Dataset Construction Process. It generates the argument structure using level-order construction until the desired number of argument forms $D$ is reached.

---

**Require:** $D$       ▷ Target number of argument forms
1: $\phi_0 \leftarrow \text{SAMPLELOGICALFORM}()$       ▷ Sample a random final conclusion
2: $a_0 \leftarrow \text{SAMPLEARGUMENTFORM}(\phi_0)$       ▷ Samples a random argument form for the final conclusion
3: $\mathcal{L} \leftarrow [a_0]$
4: $d \leftarrow 1$       ▷ Tracks the no. of argument forms
5: **while** $d < D$ **do**
6:     $P \leftarrow \text{GETALLPREMISES}(\mathcal{L})$       ▷ Extract all premises of all $a$ in $\mathcal{L}$
7:     $k \leftarrow \text{SAMPLEUNIFORM}(1, D - d)$       ▷ Sample an integer from 1 to $D - d$
8:     $P_{\text{sub}} \leftarrow \text{SAMPLERANDOMSUBSET}(P, k)$       ▷ Select $k$ premises to become subconclusions
9:     $\mathcal{L}' \leftarrow []$
10:     **for all** $\phi \in P_{\text{sub}}$ **do**
11:         $a \leftarrow \text{SAMPLEARGUMENTFORM}(\phi)$
12:         Append $a$ to $\mathcal{L}'$
13:     **end for**
14:     $\mathcal{L} \leftarrow \mathcal{L}'$
15:     $d \leftarrow d + |P_{\text{sub}}|$
16: **end while**
17: **return** Argument structure rooted at $\phi_0$

---

Table 7: Sample texts from various deductive reasoning benchmarks.

| Benchmark | Sample Text |
|---|---|
| CLUTRR [24] | Lorraine and her brother Kevin went to see a movie. Clarence took his granddaughter Lorraine to the movies and they enjoyed themselves. |
| ProofWriter [26] | The bald eagle is not rough. The bear does not need the bald eagle. The dog needs the bear. If someone is rough then they chase the bald eagle. If someone needs the bear then they are not blue... |
| ProntoQA-OOD [23] | Lempuses are bitter. Every lempus is a lorpus. Brimpuses are vumpuses. Tumpuses are impuses. Each impus is not hot. Every numpus is a sterpus. Each shumpus is brown. Sterpuses are fast. Every vumpus is not small... |
| SimpleLogic [35] | If messy and hypocritical and lonely, then shiny. If tame, then friendly. If plain and shiny and homely, then nervous. If tender, then hypocritical. If dull and impatient and plain, then tame. If spotless, then perfect. If elegant and tender, then homely... |
| LogiQA 2.0 [19] | In the past 10 years, the sales of personal notebook computers of a computer company have continued to grow, but the growth rate is lower than the growth rate of the company's total sales of all products. |
| FOLIO [12] | All people who regularly drink coffee are dependent on caffeine. People regularly drink coffee, or they don't want to be addicted to caffeine, or both. No one who doesn't want to be addicted to caffeine is unaware that caffeine is a drug... |
| JustLogic | Provided that sound travels through different kinds of matter, we know that every soul is a candidate for immortality. It is a common misconception that every soul is a candidate for immortality. |

## C    Sample texts from deductive reasoning benchmarks

Beyond metrics like vocabulary size and number of domains, the degree of natural language complexity can be straightforwardly determined by manually inspecting the linguistic patterns of a given benchmark. Table 7 shows sample texts from CLUTRR, ProofWriter, ProntoQA-OOD, SimpleLogic, LogiQA 2.0, FOLIO, and JustLogic.

Evidently, JustLogic exhibits significantly greater natural language complexity than CLUTRR, ProofWriter, ProntoQA-OOD, and SimpleLogic, because the latter benchmarks programmatically generate every sentence, while JustLogic extracts its sentences from GenericsKB, a natural language text database. Thus, the former benchmarks rely on a limited number of grammar templates, reducing their linguistic complexity. JustLogic exhibits similar levels of complexity to FOLIO. LogiQA 2.0 is more complex because it is human-curated and not backed by a formal logic system (unlike how JustLogic is backed by propositional logic). Without a formal logic system, LogiQA 2.0's argument complexity suffers, as shown in Table 3, which compromises its ability to evaluate deductive reasoning in LLMs.

## D    Prior Knowledge Independence Test

A sample prompt for the prior knowledge independence test, based on the example in Figure 1, is shown below in Figure 4. Note that the answer options vary depending on the benchmark. For example, the options for LogiQA are A, B, C, and D, while those of CLUTRR are 16 possible family relations.

## E    Experiment Implementation Details

The hyperparameters for the Llama3 models are decided largely based on the recommendations in the original paper Dubey et al. [10], which are as follows: temperature of 0.6, top p of 0.9, presence penalty of 1.15, length penalty of 1. For DeepSeek R1 and Qwen-14B (R1 Distill), the

Figure 4: Example of a prior knowledge independence test prompt.

recommended temperature of 0.6 is used. Finally, for OpenAI models, the default temperature is used. All evaluations are conducted using the OpenAI and OpenRouter APIs, with model costs ranging from \$0.0003 per question for Llama3-8B-Instruct to \$0.14 for OpenAI o1.

With regards to prompting methods, 3-shot prompting is chosen for few-shot experiments because it produces the highest accuracies compared to 6 and 9-shot. Chain-of-thought prompts also contain three examples. In the interest of fairness, all prompting techniques contain similar general instructions, which are as follows:

> You are given a paragraph of facts/premises, followed by a statement. Perform logical reasoning with propositional logic on the paragraph to determine the truth value of the statement.
>
> Here is the list of argument forms:
> - Modus Ponens
> - Modus Tollens
> - Hypothetical Syllogism
> - Disjunctive Syllogism
> - Reductio ad absurdum
> - Constructive Dilemma
> - Disjunction Elimination
>
> You must answer with either one of the 3 options:
> - TRUE: When the premises in the paragraph lead to the statement
> - FALSE: When the premises in the paragraph directly contradict the statement
> - UNCERTAIN: When the premises in the paragraph neither support nor contradict the statement
>
> Do not use your prior knowledge; your answer must be solely determined by the information within the paragraph. Assume that all premises in the paragraph are true.
>
> Question: Is the statement true, false, or uncertain?

As for the additional prompting techniques are explored in Appendix H.1, the tree-of-thought framework contains two prompts at each step: candidate generation and candidate evaluation. In addition to the general instructions above, the candidates generation prompt is shown below.

> Let's reason step by step. Generate 3 alternative possible next steps, based on the question and the answer so far. Each step consists of a single argument form, e.g. modus ponens. The question takes 1 or more steps to solve.

> Note that these 3 steps are NOT sequential. They must be alternatives to the same step.

As for candidate evaluation, the prompt is shown below. Note that the model may terminate the exploration prematurely by indicating a final answer. A practical consideration is that models tend to conclude *too early*; the prompt should be designed to emphasize exploration and instruct not to conclude unless sufficiently certain.

> Of the possible next steps, choose the one that **most directly advances the reasoning process** toward determining the truth value of the statement. Select the best next step to continue reasoning toward the answer. Do not conclude with TRUE, FALSE, or UNCERTAIN yet — unless:
>
> - All relevant reasoning paths have been explored, and
> - No further logical deduction is possible or necessary.
>
> Otherwise, output only the next reasoning step, using one valid argument form. Your goal is to build a full reasoning chain, not jump to conclusions.
>
> Only if this step logically completes the reasoning chain and no further analysis is needed, then conclude with one of: TRUE, FALSE, or UNCERTAIN.

# F  Additional Experimental Validations of the JustLogic Benchmark

## F.1  Prior Knowledge Independence Test using Other Models

To ensure that the results of the prior knowledge independence test, conducted with GPT-4 in Section 5.1, are replicable, we conduct the same test using Llama3-70B-Instruct. The results are shown in Table 8. Similar to Section 5.1, JustLogic has a high degree of prior knowledge independence, on par with other synthetically generated benchmarks, i.e. CLUTRR and ProofWriter, and substantially greater independence than the human-curated ones. Interestingly, ProofWriter's accuracy is significantly lower than random, which is potentially problematic since models may be biased *against* statements whose truth-value aligns with reality.

Table 8: Results of Prior Knowledge Independence Test using Llama3-70B-Instruct. **The lower the $|\Delta|$, the better.**

|             | $|\Delta| \downarrow$ | Accuracy (%) | Random (%) |
|-------------|-----------------------|--------------|------------|
| CLUTRR      | 5.4                   | 11.7         | 6.3        |
| ProofWriter | 8.6                   | 24.7         | 33.3       |
| LogiQA 2.0  | 23.3                  | 48.3         | 25.0       |
| FOLIO       | 10.0                  | 43.3         | 33.3       |
| JustLogic   | **6.4**               | 39.0         | 33.3       |

## F.2  Impact of Factual Accuracy on Model Performance

Given that JustLogic randomly chooses sentences from GenericsKB to add to each instance's argument structure, the final conclusion may be factually accurate or inaccurate in the real world. For example, if the conclusion is "It is not true that Japan is in Asia.", then the conclusion is factually inaccurate. Thus, there is a concern that models underperform due to confusion arising from factually inaccurate conclusions. Moreover, since some conclusions are factually accurate, such instances may exhibit artificially high performance.

To study these concerns, we conducted the following empirical study. If the above concerns are true, we expect factually inaccurate conclusions to perform worse than factually accurate ones. Because all GenericsKB sentences are factually accurate, we can straightforwardly deduce each conclusion's factual accuracy. For example, $x \lor y$ is factually accurate while $\neg x$ is not.

Figure 5 shows the comparison of accuracies for five models: DeepSeek R1, OpenAI o1-mini, GPT-4o, Llama3-70B, and Llama3-8B; the left represents when reasoning depth is 1, while the right represents when depth is 7 or less.

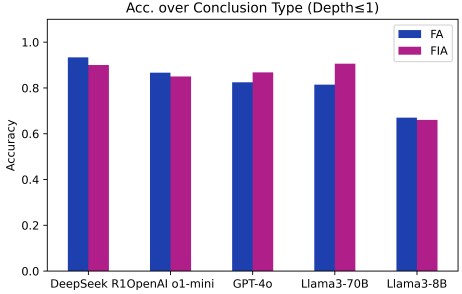 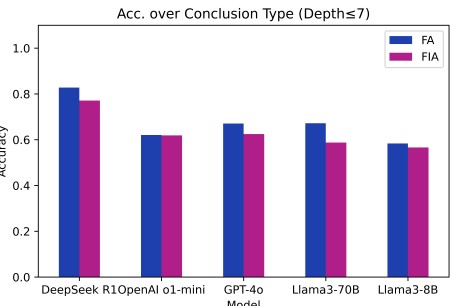

Figure 5: How factual accuracy of conclusions affects model accuracy.

These results reject the hypothesis that factually inaccurate conclusions perform worse than factually accurate ones; there is no consistent trend between both conclusion types. In fact, when depth=1, factually inaccurate conclusions exhibit higher performance for some models! At depths of 7 or less, GPT-4o and Llama3-70B saw a decrease in *relative* accuracy of factually inaccurate statements, DeepSeek R1 and Llama3-8B maintained similar accuracies, while OpenAI o1-mini saw an improvement.

There are two reasons for these results. First, our prompt explicitly instructs models to answer the question only using the paragraph provided and without using prior knowledge. The full prompt is shown in Appendix E. Moreover, in few-shot prompts, the examples provided include conclusions where their factual accuracy does not match the correct answer. These measures encourage models to ignore prior knowledge and answer questions without considering the factual accuracy of conclusions in the real world.

Second, how LLMs treat factual accuracy when reasoning deductively depends on the LLM's training: specifically, the model's ability to follow prompt instructions to ignore prior knowledge. For example, DeepSeek R1 biases toward factually inaccurate conclusions when deductively reasoning, while OpenAI o1-mini exhibits little difference in performance. Should an LLM exhibit significant differences in performance between factually accurate and inaccurate conclusions, it suggests the LLM has room for improvement in instruction following.

Importantly, the ability to deduce whether premises lead to a conclusion without using prior knowledge is a fundamental human skill: we use it to evaluate whether a debater's speech or journalist's article supports their position. The inclusion of both factually accurate and inaccurate instances in JustLogic is a feature, not a bug.

### F.3 Impact of Language "Unnaturalness" on Model Performance

Given that JustLogic is synthetically generated, there is a concern that its natural language may be highly unnatural to models, potentially hindering their ability to reason deductively. To study this concern, we compare the model perplexity of JustLogic, two other human-curated benchmarks (FOLIO and LogiQA), and two other synthetic benchmarks (CLUTRR and ProofWriter). Llama3-8B-Instruct (a non-reasoning model) and DeepSeek R1 Distill Qwen 14B (a reasoning model) are used. If JustLogic's language is indeed highly unnatural, we expect its model perplexity to be significantly higher than other benchmarks.

The results, as shown in Figure 6, reject the aforementioned hypothesis. JustLogic's model perplexities are comparable to FOLIO and lower than the rest. This shows that despite JustLogic's higher linguistic complexity (Table 3), its syntactic patterns are well understood by models. CLUTRR's and ProofWriter's higher perplexities are likely due to their unnatural symbolic-like language; LogiQA's higher perplexities are likely because its questions are originally in Chinese and did not shed their foreign syntactic patterns when translated into English. Examples can be found in Appendix C.

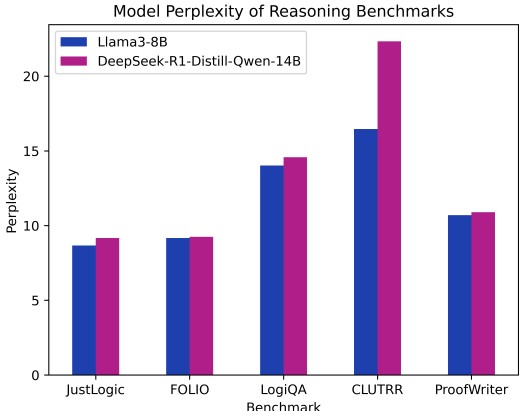

Figure 6: Model perplexities of various logical reasoning benchmarks.

Therefore, while JustLogic's natural language may seem unnatural to human readers, their syntactic patterns are highly intuitive to LLMs compared to other reasoning benchmarks. JustLogic's language likely does not hinder LLMs' understanding of the questions.

# G   Details on Human Participants

18 anonymous participants are given a random subset of questions. This is because deductive reasoning questions, especially those at high reasoning depths, are cognitively demanding and time-consuming; it is impractical to expect humans to complete 1050 questions. To ensure fairness, both models and participants are provided similar prompts and are given the same proportion of each reasoning depth. To ensure that participants understand the requirements of the task, a simple verification question is added. If they answer incorrectly, their subsequent responses are voided.

Participants are recruited from Amazon Mechanical Turk [2] and are paid $24 per hour. Participation is entirely voluntary, and the survey posed no foreseeable risks to participants. As reflected in Figure 7, to create a sample representative of the human average, the participants possess a diverse range of educational qualifications and familiarity with propositional logic.

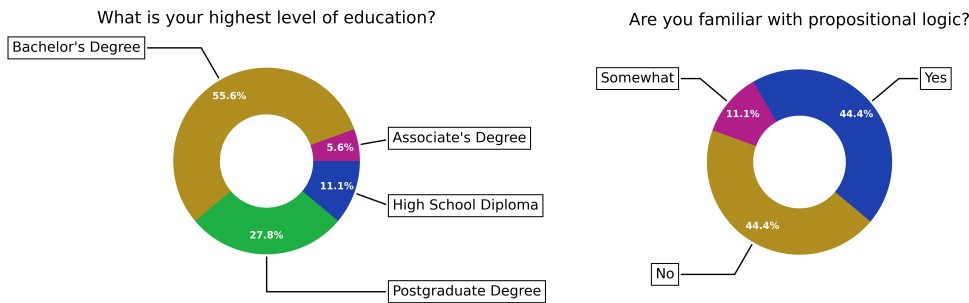

Figure 7: Participants' highest level of education and familiarity with proposition logic.

# H   Additional Model Evaluations

## H.1   Additional Evaluations on Various Prompting Techniques

While reasoning models do not require specific prompting techniques due to their reasoning-specific training, non-reasoning models observe significant deltas in accuracy based on the choice of prompts. Thus, we evaluate the best small and large non-reasoning models, Llama3-8B-Instruct and GPT-4o,

on additional prompting techniques: (i) the self-consistency decoding (SC) [28], where the answer is derived through majority voting over 5 sampled paths, and (ii) the tree-of-thought (ToT) framework [32], where each step generates 3 candidates and ultimately chooses 1; the maximum steps allowed is $depth + 2$, but the model may terminate the search earlier. Finally, we also test (iii) a CoT prompt that does not mention propositional logic. Explicit mentions of technical terms in propositional logic, e.g. reductio ad absurdum, may hinder the reasoning ability of models that are less familiar with them. This prompts tests the aforementioned hypothesis.

Table 9: Model and Human Evaluation Results.

|  | 0-shot | Few-shot | CoT | SC-CoT | ToT | CoT (w/o prop. logic) |
|---|---|---|---|---|---|---|
| Llama3-8B-Instruct | 49.8 | 41.8 | **57.8** | 54.6 | 38.6 | 54.0 |
| Llama3-70B-Instruct | 53.1 | 57.8 | **64.6** | 58.6 | 60.6 | 58.3 |
| GPT-4o-mini | 53.0 | 54.7 | **51.8** | 50.3 | 48.6 | 50.0 |
| GPT-4o | 53.8 | 58.3 | 65.6 | 67.1 | **71.4** | 67.4 |

The relative performance of the prompting techniques is heterogeneous across models. However, besides GPT-4o, we find that prompting techniques that are more expensive than vanilla CoT offer little to no performance advantage. Self-consistency CoT achieves similar performance to CoT; the former may require significantly higher sampled paths to reap its benefits. Tree-of-thought is too complex for most models to utilize, often hallucinating across prompts and failing to break down the problem into coherent steps. Lastly, we find that the explicit mention of propositional logic in the prompt is generally helpful towards model performance.

### H.2  Understanding the Performance of OpenAI o1 vs. DeepSeek R1

OpenAI o1 (72.9%) performs substantially worse than DeepSeek R1 on JustLogic, despite other benchmarks suggesting their performance should be comparable. To rule out any human errors during testing and to seek an explanation for these results, we performed a qualitative analysis of OpenAI o1's responses (all of which can be found in our GitHub repository). First, we find that o1's response to questions of depth $>= 5$ are significantly shorter than that of depth $= 3$ or $4$, which is counterintuitive. Second, o1 prematurely answers "Uncertain" for 90% of questions of depth $= 7$ without faithfully engaging with the question. Figure 8, showing OpenAI o1's and DeepSeek R1's accuracy over various difficulty levels based on argument depth, reinforces our analysis. Both models have identical accuracies for low and medium difficulty problems, but o1 struggles at high difficulty problems, performing close to random probability.

These observations suggest that OpenAI o1's test-time compute may have been artificially limited, reducing its ability to solve deep, challenging questions. Importantly, this case study reflects JustLogic's ability to flexibly probe models at various levels of difficulty.

### H.3  Futureproofing JustLogic

As LLMs improve, we expect their performance on JustLogic to rise, which necessitates increasing JustLogic's difficulty. One way is to increase the argument depth: specifically, we extended JustLogic to incorporate questions of very high depth (8 to 11), and evaluated them on the current SOTA reasoning and non-reasoning models, i.e. DeepSeek R1 and GPT-4o, using CoT prompt.

Additionally, two other benchmarks, LogiQA 2.0 and FOLIO, are also evaluated to compare their difficulty. The results, as shown in Table 10, suggest that (i) as JustLogic's question difficulty increases, model accuracy decreases, and (ii) hard JustLogic questions yield significantly lower accuracies than LogiQA 2.0 and FOLIO. This indicates that JustLogic is already more challenging than other benchmarks and is likely to remain so due to its reduced risk of performance saturation.

## I  Qualitative Analysis of Failure Modes

To identify exactly how JustLogic is challenging for existing LLMs, we conducted a qualitative analysis to identify the 4 major failure modes of various models' responses to JustLogic questions.

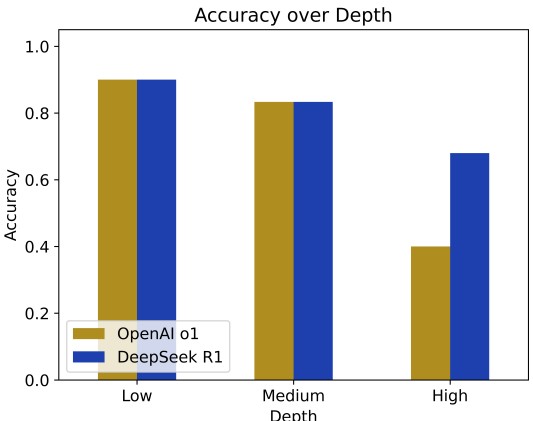

Figure 8: How reasoning depth affects accuracy for OpenAI o1 and DeepSeek R1.

Table 10: SOTA Model Performance on various JustLogic difficulty levels and other benchmarks.

| | JustLogic | | | | LogiQA 2.0 | FOLIO |
| | Easy | Medium | **Hard** | **Very Hard** | | |
|---|---|---|---|---|---|---|
| GPT-4o | 77.3 | 63.3 | **57.3** | **53.0** | 64.5 | 76.3 |
| DeepSeek R1 | 90.0 | 83.3 | **68.0** | **65.0** | 87.6 | 86.2 |

We primarily rely on analyzing the chain-of-thought responses to investigate how they produced the wrong answers. High-level explanations and abbreviated examples are provided for the failure modes.

**(1) Logical inconsistency.** Models sometimes produce arguments that rely on premises, implicit or otherwise, that contradicts earlier parts of their chain-of-thought. This causes models to generate incoherent arguments, and ultimately arrive at the wrong conclusion. For example, the following is a truncated response from OpenAI o1:

```
While R implies that researchers are observing data,
without specific information, we cannot definitively
state R → S.
...
Since M ∨ R is true and M ∨ S is the statement we
are evaluating, and we have established M is true if
N is true, we can conclude:
 • M ∨ S is true.
```

This line of reasoning suggests that $M \vee R$ implies $M \vee S$. For this to be the case, $R \to S$ must be true. However, earlier in the reasoning chain, o1 says "we cannot definitively state $R \to S$". This argument is therefore logically inconsistent.

**(2) Wrong application of argument forms.** Models sometimes identify the correct argument form to solve the question. However, mistakes are made when applying the form to the specific question context. In the truncated response below by Llama3-70B, disjunctive syllogism is, in fact, the appropriate argument form, but the opposite conclusion should have been made: the statement, pain can be severe and make a person unable to perform normal activities, does not hold.

```
Premise 1: Either condensation is water vapor
changing to liquid water or the statement, pain
can be severe and make a person unable to perform
normal activities, does not hold.
```

```
Premise 2:  The claim that condensation is water
vapor changing to liquid water does not correspond
with reality.

Using disjunctive syllogism with premises 1 and
2, we conclude that the statement, pain can be
severe and make a person unable to perform normal
activities, holds.
```

**(3) Using the wrong argument form.** Sometimes, models attempt to use argument forms that logically cannot be applied to the context. Such mistakes often derail the response, making it entirely incoherent. In the following example of an output by Llama3-70B, modus tollens simply cannot be applied here; reductio ad absurdum should have been used instead.

```
Premise 1:  One may reasonably assume that if the
statement that 'football is a memory-making machine'
is incorrect, then fairways are lawns.  Premise
2:  Assuming the statement that 'football is a
memory-making machine' is incorrect, we know that
it is not the case that fairways are lawns.

Using modus tollens with premise 1 and 2...
```

**(4) False interpretation of facts.** Models sometimes misinterpret the natural language facts entirely. This is most clearly seen when models translate sentences into the wrong logical form. While some of these translations seem trivially simple, especially for LLMs, such mistakes are as common as the other failure modes. In the example below, GPT-4o's interpretation of Premise 2 is incorrect: Premise 2 does in fact imply that the statement "most roses grow fairly rapidly" is false because of the word "mistakenly."

```
Premise 2:  Some people mistakenly believe that most
roses grow fairly rapidly.

We cannot directly use Premise 2 to confirm or deny
(¬Q), as it only mentions a mistaken belief rather
than the truth value.  Therefore, we do not have
sufficient information to directly negate (Q).
```

Nonetheless, some sentences are more complex and therefore more prone to false interpretations. In the example below, also by GPT-4o, Premise 2's logical form should be $\neg(A \to B) \to \neg C$ instead.

```
Premises:
...
2.  "Given that the claim that if police sergeants
receive calls, then good nutrition helps reduce
low birth weight, miscarriage and anemia does not
reflect reality, it can be inferred that some people
mistakenly believe that oil is simply a liquid form
of fat."

From Premise 2:  (¬(A→B))
```

## J  Limitations & Future Works

While JustLogic already achieves higher or similar natural language complexity to existing deductive reasoning benchmarks, as shown in Section 3.4, linguistic complexity can be further enhanced to emulate human-written prose, e.g. news articles and fiction stories. Notably, LLMs can be introduced in Step 2 of JustLogic's dataset construction process, whereby instead of randomly selecting sentences

from GenericsKB, an LLM can generate fictional statements and scenarios, e.g. "John's favorite food is hamburgers.". While LLM generation has been successful in datasets involving inductive reasoning and commonsense knowledge, e.g. MuSR [25], it is currently too unreliable for deductive reasoning due to several common mistakes, e.g. ignoring instructions, hallucination, and invalid logic. Nonetheless, as LLMs become more reliable, LLM generation is a promising approach worthy of further exploration.

Error analysis using JustLogic can also be further explored. Interesting research questions include: Are models able to use argument forms appropriately? At which step of the argument chain does the model usually fail? What are the most common reasons for failure? These insights may be useful for fine-tuning models for logical reasoning tasks [20] and model guidance [3].

JustLogic can be scaled to incorporate more question types related to logical reasoning, such as multiple-choice questions, identifying missing premises in arguments, identifying logical fallacies in arguments, and natural language sentence to formal logic translation. [20] provides a comprehensive taxonomy. JustLogic's program can be adapted to accommodate each question type while maintaining its key advantages. By measuring deductive reasoning across multiple modalities using a single dataset construction method, JustLogic can provide more comprehensive and controlled evaluations and error analysis.

