# OpenReview forum: "JustLogic: A Comprehensive Benchmark for Evaluating Deductive Reasoning in LLMs"
_NeurIPS.cc/2025/Datasets_and_Benchmarks_Track — Submitted to NeurIPS 2025 Datasets and Benchmarks Track_

### Official Review · Reviewer_DTBj · 2025-06-01

**Rating:** 5
**Confidence:** 5

**Summary:**

This work introduces *JustLogic*, a deductive reasoning benchmark for large language models LLMs. The benchmark is code-generated by first sampling an underlying reasoning structure, allowing for the theoretical generation of an infinite number of instances. Each meta-argument is then sampled from a diverse pool of statements, which is intentionally designed to diverge from real-world knowledge, preventing LLMs from solving tasks using memorized factual information. Since all instances are generated programmatically, the benchmark also enables in-depth error analysis.

**Dataset Code Accessibility:**

Yes

**Dataset Code Comments:**

https://huggingface.co/datasets/michaelchenkj/JustLogic/viewer/default/train?row=7&views%5B%5D=train

**Ethical Considerations:**

No, there are no or only very minor ethics concerns

**Final Justification:**

My concerns in the initial review have been almost addressed.

**Limitations Weaknesses:**

While the authors claim that JustLogic improves upon prior benchmarks in terms of natural language complexity (how arguments are linguistically expressed), I believe that it still lacks diversity in how logical forms are represented.

- **Structural limitations:** The benchmark omits some important logical constructs such as *exclusive OR (XOR)* and *conjunctive AND*, which are commonly encountered in real-world reasoning tasks.
- **Linguistic variation:** Each logical relationship is expressed using only a limited set of surface forms. In real-world language, such relationships can be conveyed in a much more diverse and nuanced manner, and are often communicated implicitly rather than through explicit phrasing.

**Strengths Contributions:**

+ I personally appreciate the use of code to generate the benchmark, as it offers several advantages: (1) it allows for the generation of an effectively infinite number of instances; (2) it enables precise control over instance properties (e.g., reasoning depth), which is valuable for adjusting difficulty and conducting targeted analysis; and (3) as the authors note, if the JustLogic test set were ever leaked, a new set of comparable difficulty could be easily regenerated—extending the benchmark’s practical lifecycle.
+ The decision to include factually inaccurate statements in the instances is also thoughtful and well-motivated, as it prevents LLMs from relying on memorized knowledge and ensures that the task truly tests reasoning rather than recall.

---

> ### Author Rebuttal · Authors · 2025-07-27
>
> Thank you for your detailed and constructive review. We are glad that you appreciate the use of code generation and the novel dataset construction method. We have carefully taken into consideration your concerns regarding complexity/diversity, and have substantially improved our paper based on your feedback.
>
> **C1**: *Structural limitations: The benchmark omits some important logical constructs such as exclusive OR (XOR) and conjunctive AND, which are commonly encountered in real-world reasoning tasks.*
>
> Thank you for highlighting this concern. It is indeed important that these constructs are reflected in JustLogic’s generated logical structures. JustLogic only explicitly defines the four fundamental logical forms for the sake of simplicity. But importantly, all other logical forms, including exclusive or (⊕) and conjunction (∧), can be constructed using the four basic forms. Given that JustLogic’s program can express these constructs, we posit that JustLogic does not have such structural limitations.
>
> To directly address this, we have added a new Appendix section as follows:
>
> > For the sake of simplicity, JustLogic explicitly defines the most fundamental logical forms only: basic ($x$), negation ($¬x$), conditional ($x → y$), and disjunction ($x ∨ y$). This is because all other logical forms, including exclusive or ($⊕$) and conjunction ($∧$), can be constructed by the program.
> >
> >&nbsp;
> >
> > As a case study, we show how the four basic forms construct the conjunction ($x ∧ y$), and provide evidence of its existence in the dataset, despite never being explicitly defined. Using De Morgan’s Law, the following logical equivalence can be derived:
> >
> > $x ∧ y  ≡  ¬(¬x ∨ ¬y)$
> >
> > This can be further transformed by replacing the disjunction with implication and negation:
> >
> > $x ∧ y  ≡  ¬(¬x ∨ ¬y)  ≡  ¬(x → ¬y)$
> >
> >&nbsp;
> >
> > Indeed, the final transformation is found in the actual dataset. For example, the following is a Modus Tollens argument structure containing the above logical form, extracted from the training dataset generated organically by JustLogic:
> >
> >&nbsp;
> >
> > $¬(a → ¬b) → c $
> >
> > $¬ c$
> >
> > $∴ a → ¬b$
> >
> >&nbsp;
> >
> > The JustLogic construction program evidently can and does generate logical forms beyond the four explicitly defined ones. Nonetheless, we note that the natural language translation will express the given logical form as such. In future works, we intend to adapt the translation program such that such forms will be expressed in their simplified versions. In the first premise of the above example, the natural language text is: *When the claim that "if an editorial is an article, then it is not the case that players have rights” has no merit is true, it follows that antibiotics are drugs that kill or impair bacteria*. In future works, it will be adapted to read: *If an editorial is an article and players have rights, then it follows that antibiotics are drugs that kill or impair bacteria.*
>
> **C2**: *Linguistic variation: Each logical relationship is expressed using only a limited set of surface forms. In real-world language, such relationships can be conveyed in a much more diverse and nuanced manner, and are often communicated implicitly rather than through explicit phrasing.*
>
> Thank you for your insightful feedback; your point on linguistic variation is well-taken.
>
> Regarding the limited set of surface forms, this is true given that the four basic forms have a combined 50 templates. However, given that many basic forms are combined to form an argument, the space of possible expressions is enormous. For example, just 2 basic forms and 2 conditionals lead to 30,976 variations! Besides, this does not include the considerable linguistic diversity of the randomly selected sentences from the GenericsKB database.
>
> Further, we note that the number of templates can be trivially expanded if needed. To showcase this, we release an expanded database with 200 templates, which are first synthetically generated and then manually verified. To comply wth the NeurIPS rebuttal policy, this database will be released after the rebuttal period.
>
> Regarding linguistic expressions that are more “nuanced” and “communicated implicitly”, these are indeed interesting research directions. Can implicit phrasings be formalized and incorporated into a natural language translation program? Are LLMs advanced enough to generate natural language with implicit semantic meanings, given a symbolic logical structure?
>
> Given that these questions are still under active investigation in the literature, they are beyond the scope of this work. We have adapted the first paragraph of Limitations & Future Works to reflect your feedback:
>
> > While JustLogic already achieves higher or similar natural language complexity to existing deductive reasoning benchmarks, as shown in Section 3.4, linguistic complexity can be further enhanced to emulate human-written prose, e.g. news articles and fiction stories. Specifically, real-world conversations sometimes convey logical relations implicitly. For example, instead of saying “If you miss the deadline, then you will be fired.”, one may say “Miss the deadline, and you’re fired.”. Formalizing such nuanced statements has been a topic of interest for the NLP community; advances in this area will enhance JustLogic’s natural language. Further, with the recent explosion of LLM-generated datasets, LLM could be introduced in Step 2 of JustLogic's dataset construction process, whereby instead of randomly selecting sentences from GenericsKB, an LLM can generate fictional statements and scenarios. While LLM generation has found success in inductive reasoning and commonsense knowledge datasets, e.g. MuSR [25], it is currently too unreliable for deductive reasoning due to fatal problems such as ignoring instructions, hallucination, and invalid logic. Nonetheless, as LLMs become more reliable, LLM generation is a promising approach worthy of further exploration.

---

> > ### Comment · Reviewer_DTBj · 2025-08-01
> >
> > Thanks for your rebuttal. I am happy to increase the score from 4 to 5

---

> > > ### Author Response · Authors · 2025-08-02
> > > **Response by Authors**
> > >
> > > Thank you for your positive feedback. We are glad that our response fully addressed your comments.

---

### Official Review · Reviewer_1yrT · 2025-06-28

**Ethics Flags:** Data quality and representativeness
**Rating:** 5
**Confidence:** 3

**Summary:**

This paper focuses on evaluating the deductive reasoning capabilities of Large Language Models (LLMs). Existing deductive reasoning benchmarks suffer from the lack of task complexity, the presence of prior knowledge as a confounder, and superficial error analysis. To this end, the authors introduce JustLogic, a synthetically generated benchmark designed for rigorous evaluation of LLMs. It is 1) high complexity, 2) prior knowledge independent, and 3) capable of in-depth error analysis. Specifically, they first generate an argument structure with the pre-defined four logical forms. Then they add nature language statements to the argument structure and generate a query statement to form the dataset. Eventually, they conducted comprehensive experiments on the JustLogic.

**Dataset Code Accessibility:**

Yes

**Ethical Considerations:**

No, there are no or only very minor ethics concerns

**Final Justification:**

I've read the authors' rebuttal and will keep my rating.

**Limitations Weaknesses:**

1.	Can you analyze more towards the thinking process of the LLMs? I think the paper would be better if the authors could propose some metrics to evaluate the deductive reasoning process of the LLMs.

2.	Can a similar construction pipeline be applied to the multi-modal field?

**Strengths Contributions:**

1.	The way to construct JustLogic is reasonable considering the structure of the deductive reasoning data is strictly built upon the four types of expressions of logical forms. Then they add natural language statements to the argument structure. This is reasonable for decoupling the prior knowledge and controlling the complexity of the generated arguments.

2.	Compared with the existing deductive reasoning benchmark, JustLogic is indeed distinct due to its large scale and reasoning depth.

3.	The authors conduct comprehensive experiments on the JustLogic dataset and bring in some insights to the observation.

---

> ### Author Rebuttal · Authors · 2025-07-27
>
> Thank you for your insightful and constructive review. We are grateful for your positive assessment of JustLogic’s novelty and the provided insights. Following your suggestions, we have made substantial revisions to our paper.
>
> **C1**: *Can you analyze more towards the thinking process of the LLMs? I think the paper would be better if the authors could propose some metrics to evaluate the deductive reasoning process of the LLMs.*
>
> Thank you for offering this valuable perspective. Indeed, in future iterations of JustLogic, we intend to incorporate evaluation processes and metrics that evaluate the LLMs’ chain-of-thought processes, rather than just the final answer.
>
> We did not incorporate these metrics into the current benchmark due to the significant difficulty imposed by the unstructured nature of chain-of-thought outputs. There is no straightforward implementation beyond manually reading the LLM outputs, which is expensive and unscalable. Nonetheless, as a preliminary investigation, we propose an evaluation pipeline and a series of metrics, which are described in a newly added Appendix (“Towards a quantitative investigation of LLMs’ chain-of-thought”) section as follows:
>
> > Reasoning benchmarks typically evaluate LLMs by measuring final answer accuracy; in our case, we evaluate whether LLMs derive the correct conclusion: true, false, or uncertain. However, it is also helpful to understand the performance of the chain-of-thought itself. JustLogic’s dataset construction process lays the groundwork for such an evaluation: the argument structure is synthetically generated, so we can automatically and cleanly delineate each step in the argument. As a first step, we propose a simple evaluation pipeline and a series of metrics to investigate the quality of the reasoning process, inspired by MME-CoT [37].
> >
> >&nbsp;
> >
> > First, the LLM CoT responses must be segmented into a list of explicit steps. This can be automatically done by prompting an LLM (henceforth “Evaluation LLM”). The ground truth reasoning process, as generated by the JustLogic construction program, must also be converted into a list of steps in a similar format. This can be trivially achieved with some data processing.
> >
> >&nbsp;
> >
> > Second, the LLM reasoning steps are compared against the ground truth steps, leaving aside the predicted answer. Similar to the previous step, the Evaluation LLM is employed as the primary method of comparison. Two possible metrics are:
> >
> >&nbsp;
> >
> > - **Precision**: The proportion of LLM reasoning steps that are logically correct and relevant towards deriving the final answer.
> >
> > - **Recall**: The proportion of ground truth steps that appear in the LLM reasoning steps.
> >
> >&nbsp;
> >
> > As an example, consider a JustLogic question where the four required steps $a → b → c → d$ lead to the final answer $X$. The table below shows 3 hypothetical model responses and their precision, recall, and final answer accuracy scores. Precision and recall can provide valuable information about model performance: they penalize responses with correct final answers but wrong/incomplete reasoning processes (Examples 1 & 2), while rewarding those with wrong final answers but correct reasoning traces (Example 3).
> >
> >&nbsp;
> >
> > |                         | Accuracy | Precision | Recall |
> > |-------------------------|:--------:|:---------:|:------:|
> > | $b → c = X$               | 1        | 1.0       | 0.5    |
> > | $e → a → b → c → d → f = X$ | 1        | 0.67      | 1.0    |
> > | $a → b → c → d = Z$       | 0        | 1.0       | 1.0    |
> >
> >&nbsp;
> >
> > Notably, tackling the unstructured nature of CoT outputs is a significant challenge. Specifically, how accurately does the Evaluation LLM measure precision and recall? Since LLM reasoning steps often diverge significantly in wording from the ground truth, there is a significant risk of false positives and negatives. Further, how do we account for any potential biases in the chosen model for Evaluation LLM? For example, GPT-4o, as the Evaluation LLM, may judge GPT-4o mini more favourably due to stylistic and shared knowledge biases. These pertinent questions have not been resolved in the literature.
> >
> >&nbsp;
> >
> > Given the need for substantial human evaluations and ablation studies, this research direction is beyond the scope of this work and deserves a separate study. Nonetheless, we hope that the JustLogic dataset and this initial exploration lay the groundwork for future research into LLMs’ thinking processes.
>
> In future iterations of JustLogic, we plan to focus on implementing robust evaluations of LLM CoT responses.
>
> **C2**: *Can a similar construction pipeline be applied to the multi-modal field?*
>
> Thank you for this valuable suggestion. Introducing multi-modality to JustLogic is indeed feasible; JustLogic’s dataset construction process can be adapted to accommodate alternative modalities. We have added your suggestion to Appendix J (Limitations & Future Works) to encourage further research in this direction:
>
> > JustLogic is currently an English, text-based dataset. However, importantly, it has enormous potential to be multilingual and multimodal…It is also entirely feasible to make the dataset multimodal. As a straightforward example, symbols can be used to reference other modalities, rather than just being replaced by text. For example, “If x, then y.” can be replaced by “If the country in the map is shaded red, then it is currently in a recession.”. JustLogic’s extensibility and flexibility create immense, exciting opportunities for future research.
>
> Nonetheless, we note that such an implementation, while meaningful, is likely a nontrivial challenge. For example, the phrase “the country in the map is shaded red” must be accompanied by an image with these characteristics. Creating or finding a suitable multimodal dataset for this purpose, and integrating it into JustLogic, will be a substantial undertaking.
>
> [37] Jiang, D., Zhang, R., Guo, Z., Li, Y., Qi, Y., Chen, X., ... & Li, H. (2025). Mme-cot: Benchmarking chain-of-thought in large multimodal models for reasoning quality, robustness, and efficiency. arXiv preprint arXiv:2502.09621.

---

### Official Review · Reviewer_ZEcP · 2025-07-01

**Rating:** 5
**Confidence:** 4

**Summary:**

This paper introduces a new benchmark dataset, JustLogic, for evaluating deductive reasoning in natural language. The authors address three major shortcomings in existing reasoning benchmarks: (1) insufficient complexity—both in natural language and argument structure; (2) vulnerability to prior knowledge, which hinders the isolation of true deductive reasoning; and (3) a lack of in-depth error analysis. JustLogic programmatically generates argument structures and incorporates natural language via factually inaccurate but realistic sentences, balancing linguistic diversity and prior knowledge independence. The paper includes thorough experiments demonstrating that current SOTA LLMs significantly underperform humans on this benchmark and provides detailed error analysis.

**Additional Feedback:**

- Consider adding a small-scale fine-tuning experiment to assess whether LLMs can learn deductive forms from a few JustLogic examples.

- Include a full worked example in the appendix, showing the full process from argument structure to natural language instantiation, query generation, model prediction, and correct answer.

- Future work could explore multilingual versions of JustLogic to evaluate cross-lingual deductive reasoning abilities.

**Dataset Code Accessibility:**

Yes

**Ethical Considerations:**

No, there are no or only very minor ethics concerns

**Limitations Weaknesses:**

1. While the use of factually incorrect sentences aids in removing prior knowledge bias, it may reduce the relevance of the benchmark to real-world tasks (e.g., QA, decision support). A discussion on bridging synthetic and real-world tasks would be helpful.

2. The impact of the preamble and prompt design on model performance is acknowledged but not deeply analyzed. More ablation studies or control experiments would strengthen the evaluation.

**Strengths Contributions:**

1. The paper clearly identifies key limitations in existing benchmarks—especially the trade-off between natural language and argument complexity—and addresses a pressing need for a more robust evaluation of LLMs’ deductive reasoning abilities.

2. JustLogic’s programmatic generation of diverse argument structures, combined with factually inaccurate but realistic natural language from real-world sources, strikes a smart balance between control and naturalism. Its design also allows for easy scaling and future-proofing.

3. The paper provides a comprehensive set of experiments, including prior knowledge independence testing, human comparisons, and in-depth error analysis across reasoning depths and argument forms. The open-source implementation further enhances reproducibility and community adoption.

---

> ### Author Rebuttal · Authors · 2025-07-27
>
> Thank you for your detailed and constructive reviews. We’re glad that you appreciate the limitations of existing reasoning benchmarks and how JustLogic addresses them. We have revised our paper based on your feedback.
>
> **C1**: *While the use of factually incorrect sentences aids in removing prior knowledge bias, it may reduce the relevance of the benchmark to real-world tasks (e.g., QA, decision support). A discussion on bridging synthetic and real-world tasks would be helpful.*
>
> Thank you for this valuable suggestion. It is indeed important to discuss how JustLogic, a synthetic task, is highly relevant and crucial for evaluating LLMs’ performance in real-world reasoning scenarios. We add the following text to our paper:
>
> > In real-world reasoning scenarios, the most fundamental skill required is deductive reasoning; all other skills, such as background knowledge and instruction following, serve a supporting, albeit important, role. To advance SOTA LLM capabilities, researchers must diagnose whether their models are limited by their deductive reasoning ability, which must be evaluated in isolation. Against the backdrop of existing reasoning benchmarks that are confounded by the use of prior knowledge and other factors, JustLogic uniquely fulfills this crucial function. In practice, we recommend LLM researchers to utilize JustLogic as a key pillar in a broader suite of benchmarks that encompasses the range of skills required for real-world reasoning scenarios.
>
> **C2**: *The impact of the preamble and prompt design on model performance is acknowledged but not deeply analyzed. More ablation studies or control experiments would strengthen the evaluation.*
>
> Thank you for this insightful point. We agree that ablation studies are important to control for the effects of various prompt designs. Appendix H.1 compares the 6 different prompt designs across 4 models, finding that CoT prompts usually perform the best, even as compared to more complex prompting techniques. For your convenience, the following is the contents of the section:
>
> > While reasoning models do not require specific prompting techniques due to their reasoning-specific training, non-reasoning models observe significant deltas in accuracy based on the choice of prompts. Thus, we evaluate the best small and large non-reasoning models, Llama3-8B-Instruct and GPT-4o, on additional prompting techniques: (i) the self-consistency decoding (SC) [28], where the answer is derived through majority voting over 5 sampled paths, and (ii) the tree-of-thought (ToT) framework [32], where each step generates 3 candidates and ultimately chooses 1; the maximum steps allowed is $depth+2$, but the model may terminate the search earlier. Finally, we also test (iii) a CoT prompt that does not mention propositional logic. Explicit mentions of technical terms in propositional logic, e.g. reductio ad absurdum, may hinder the reasoning ability of models that are less familiar with them. This prompts tests the aforementioned hypothesis.
> >
> >&nbsp;
> >
> > | Model               | 0-shot | Few-shot | CoT   | SC-CoT | ToT   | CoT (w/o prop. logic) |
> > |---------------------|:------:|:---------:|:------:|:-------:|:------:|:----------------------:|
> > | Llama3-8B-Instruct  | 49.8  | 41.8     | **57.8** | 54.6   | 38.6  | 54.0                 |
> > | Llama3-70B-Instruct | 53.1  | 57.8     | **64.6** | 58.6   | 60.6  | 58.3                 |
> > | GPT-4o-mini         | 53.0  | 54.7     | **51.8** | 50.3   | 48.6  | 50.0                 |
> > | GPT-4o              | 53.8  | 58.3     | 65.6  | 67.1   | **71.4** | 67.4                 |
> >
> >&nbsp;
> >
> > The relative performance of the prompting techniques is heterogeneous across models. However, besides GPT-4o, we find that prompting techniques that are more expensive than vanilla CoT offer little to no performance advantage. Self-consistency CoT achieves similar performance to CoT; the former may require significantly higher sampled paths to reap its benefits. Tree-of-thought is too complex for most models to utilize, often hallucinating across prompts and failing to break down the problem into coherent steps. Lastly, we find that the explicit mention of propositional logic in the prompt is generally helpful towards model performance.
>
> **C3**: *Consider adding a small-scale fine-tuning experiment to assess whether LLMs can learn deductive forms from a few JustLogic examples.*
>
> Thank you for your insightful feedback. As you’ve suggested, we conducted several small-scale fine-tunes to study whether LLMs can learn from just a few JustLogic examples. We train Llama3-8B-Instruct using LoRA and QLoRA fine-tuning on a subset of the JustLogic training dataset (2100 examples with depth <= 3). The hyperparameters are: learning rate of 1e05, batch size of 32, 2 epochs, LoRA rank of 16, LoRA alpha of 32, LoRA dropout of 0.10.
>
> The results are as follows: 57.8% for Llama3-8B without any fine-tuning, 64.9% for QLoRA, and 67.0% for LoRA. These results indicate that LLMs can indeed improve their deductive reasoning capabilities by fine-tuning on JustLogic examples. This presents a further use case for JustLogic beyond evaluation: JustLogic is also capable of enhancing LLM performance.
>
> **C4**: *Include a full worked example in the appendix, showing the full process from argument structure to natural language instantiation, query generation, model prediction, and correct answer.*
>
> Thank you for this valuable suggestion. We will add a full worked example in the form of a figure: it will first show how the question is constructed, in a similar format to Figure 2, followed by the model response and final answer generated by GPT-4o. Due to the inability to attach a figure, we write the contents below as text, which will be converted into a figure in the actual paper:
>
> > **Step 1: Generate the argument structure**
> >
> > Using the disjunction elimination argument form:
> >
> > $a ∨ b$ (Premise 1)
> >
> > $a → c$ (Premise 2)
> >
> > $b → c$ (Premise 3)
> >
> > $∴ c$
> >
> >&nbsp;
> >
> > Then, using modus ponens:
> >
> > $c → d$ (Premise 4)
> >
> > $c$ (Derived from the above subargument)
> >
> > $∴ d$ (Final conclusion)
>
> > **Step 2: Add natural language sentences to the argument**
> >
> > First, the symbols ($a$ to $d$) are replaced with randomly selected sentences from the GenericsKB database.
> >
> > - $a$: Leadership is a self-referral process.
> >
> > - $b$: Cats love a cozy, enclosed space to curl up in at nap time.
> >
> > - $c$: Size is a function of the ability to conquer space.
> >
> > - $d$: Most cobras possess deadly venom.
> >
> > &nbsp;
> >
> > Second, the logical connectives are translated into natural language using the logical form expression templates.
> >
> > - Premise 1: There is good reason to believe that either leadership is a self-referral process or cats love a cozy, enclosed space to curl up in at nap time.
> >
> > - Premise 2: Whenever it is true that leadership is a self-referral process, 'size is a function of the ability to conquer space' is true.
> >
> > - Premise 3: When cats love a cozy, enclosed space to curl up in at nap time is true, it follows that size is a function of the ability to conquer space.
> >
> > - Premise 4: Once we know size is a function of the ability to conquer space, we also know that most cobras possess deadly venom.
>
> > **Step 3: Putting it together**
> >
> > Finally, these premises are strung together to form a paragraph, which is provided to models as context to answer the question. In this case, we ask the LLM whether this statement is true: “It is a simple truth that most cobras possess deadly venom.”. The answer is true.
>
> > **Sample model response**. The following is GPT-4o’s answer to this question, where it arrived at the correct answer:
> >
> > *Not shown due to rebuttal character limit*
>
> **C5**: *Future work could explore multilingual versions of JustLogic to evaluate cross-lingual deductive reasoning abilities.*
>
> Thank you for highlighting this. Creating multilingual versions of JustLogic is indeed an exciting research direction. We have added your suggestion to Appendix J (Limitations & Future Works) as a direction for our future work:
>
> > JustLogic is currently an English, text-based dataset. However, importantly, it has enormous potential to be multilingual and multimodal. To make the dataset multilingual, researchers need only replace the natural language sentences used in Step 2 of the dataset construction process, i.e., replace the list of logical form expression templates and GenericsKB database. Otherwise, the dataset construction process is generalizable to any language…JustLogic’s extensibility and flexibility create immense, exciting opportunities for future research.

---

### Official Review · Reviewer_LztV · 2025-07-03

**Rating:** 5
**Confidence:** 4

**Summary:**

The paper introduces JustLogic, a synthetic benchmark designed to evaluate deductive reasoning in large language models (LLMs). The authors argue that existing benchmarks suffer from key limitations: insufficient complexity, dependence on prior knowledge, and lack of detailed error analysis. JustLogic addresses these by 1) generating complex, varied logical problems, 2) eliminating prior knowledge biases, and 3) enabling fine-grained analysis of errors based on reasoning depth and argument structure. Experiments show that reasoning-augmented LLMs perform near the human average but below the human ceiling, while standard models lag further behind.

**Dataset Code Accessibility:**

Yes

**Ethical Considerations:**

No, there are no or only very minor ethics concerns

**Final Justification:**

I've read the authors' rebuttal and will keep my rating.

**Limitations Weaknesses:**

* While synthetic data generation removes biases, it may not fully reflect real-world reasoning scenarios where background knowledge and context matter.
* Small MT sample size may not robustly represent the "human average"

**Strengths Contributions:**

* Design the benchmark to ensure the evaluations focusing purely on deductive reasoning rather than memorization or prior knowledge
* Generate a wide range of linguistic patterns, vocabulary, and argument structures, making it more challenging
* Allow researchers to do more fine-grained error analysis on reasoning depth and argument structure

---

> ### Author Rebuttal · Authors · 2025-07-27
>
> Thank you for your thoughtful review. We are glad that you recognize JustLogic’s various advantages, including its adjustable level of difficulty and fine-grained error analysis.
>
> **C1**: *While synthetic data generation removes biases, it may not fully reflect real-world reasoning scenarios where background knowledge and context matter.*
>
> Thank you for providing this valuable feedback. We clarify that JustLogic is not designed to address the entire spectrum of challenges in real-world reasoning scenarios. Instead, JustLogic’s purpose is to (i) allow researchers to evaluate LLM deductive reasoning ability in isolation, given its critical importance, and (ii) serve as a key pillar in a broader suite of benchmarks that encompasses the range of skills required. To better motivate this purpose, we add the following paragraph to our paper:
>
> > In real-world reasoning scenarios, the most fundamental skill required is deductive reasoning; all other skills, such as background knowledge and instruction following, serve a supporting, albeit important, role. To advance SOTA LLM capabilities, researchers must diagnose whether their models are limited by their deductive reasoning ability, which must be evaluated **in isolation**. Against the backdrop of existing reasoning benchmarks that are confounded by the use of prior knowledge and other factors, JustLogic uniquely fulfills this crucial function. In practice, we recommend LLM researchers to utilize JustLogic as a key pillar in a broader suite of benchmarks that encompasses the range of skills required for real-world reasoning scenarios.
>
> **C2**: *Small MT sample size may not robustly represent the "human average"*
>
> Thank you for raising this concern. We agree that our sample size does not fully encapsulate the human average. After all, a robust estimate will require a population across all ages, education levels, areas of expertise, etc, which cannot feasibly be conducted in this study. To reflect your feedback, we qualify the strength of our human evaluation results by adding the following to the paper:
>
> > We note that the human evaluation results may not be fully reflective of the range of human abilities across different ages, educational levels, areas of expertise, etc, especially given the small sample size. Rather than representing conclusive measurements, our results should be interpreted as indicative guidelines.

---

### Decision · Program_Chairs · 2025-09-18

**Decision:**

Reject

**Comment:**

This paper presents a dataset for deductive reasoning in LLMs with higher natural language complexity than related datasets like CLUTRR. The dataset consists of synthetically generated reasoning chains that are turned into natural language via GenericsKB.

Strengths:
1. Carefully constructed dataset with biases removed compared to prior datasets
2. Compared thoroughly to prior dataset efforts

Weaknesses
1. No engagement with background knowledge. This makes the dataset synthetic and and not as appropriate for modeling real-world deductive reasoning scenarios.
2. Lack of a strong reason for this dataset to be produced. It cleans up prior datasets but I don't see it as a substantial advancement of the field.

Discussion: one point that came up is the potential to include analysis of the thinking process. The authors argue this is fundamentally hard to do.

Overall, this paper produces a new dataset which cleans up some issues from prior dataset efforts. The quality of the dataset itself is good. However, I don't see a strong reason why this dataset is needed now. Reasoning-equipped language models from the o1 generation are already performing on par with average humans. I understand there is still headroom, but it's not clear to me that squeezing out this remaining headroom will actually help on any problems of real-world importance, given the artificial setting. It's also not clear to me that these past datasets aren't reasonable barometers of this progress, despite their issues or potential spurious correlations.

More broadly, such isolated tests of deductive reasoning (of which there are several, detracting from the novelty of this paper) were needed a couple of years ago when capabilities of models were more nascent and not well-understood. In an environment where LLMs are solving IMO problems, this dataset appears to miss the mark a bit in terms of where the forefront of evaluation is.

Ultimately, a great many papers have scores in the "accept" range in this track. I don't think this dataset is one that is of high importance to be published among the possible candidates. This decision reflects discussion with the SAC about this paper as well.